# Design of Power-Efficient Training Accelerator for Convolution Neural Networks

JiUn Hong [1], Saad Arslan [2], TaeGeon Lee [1] and HyungWon Kim [1,*]

1 Department of Electronics Engineering, Chungbuk National University, Chungdae-ro 1, Seowon-gu, Cheongju 28644, Korea; hongju@chungbuk.ac.kr (J.H.); greet921113@gmail.com (T.L.)
2 Department of Electrical and Computer Engineering, COMSATS University Islamabad, Park Road, Tarlai Kalan, Islamabad 45550, Pakistan; saad.arslan@comsats.edu.pk
* Correspondence: hwkim@chungbuk.ac.kr

**Abstract:** To realize deep learning techniques, a type of deep neural network (DNN) called a convolutional neural networks (CNN) is among the most widely used models aimed at image recognition applications. However, there is growing demand for light-weight and low-power neural network accelerators, not only for inference but also for training process. In this paper, we propose a training accelerator that provides low power and compact chip size targeted for mobile and edge computing applications. It accelerates to achieve the real-time processing of both inference and training using concurrent floating-point data paths. The proposed accelerator can be externally controlled and employs resource sharing and an integrated convolution-pooling block to achieve low area and low energy consumption. We implemented the proposed training accelerator in an FPGA (Field Programmable Gate Array) and evaluated its training performance using an MNIST CNN example in comparison with a PC with GPU (Graphics Processing Unit). While both methods achieved a similar training accuracy of 95.1%, the proposed accelerator, when implemented in a silicon chip, reduced the energy consumption by 480 times compared to the counterpart. Additionally, when implemented on an FPGA, an energy reduction of over 4.5 times was achieved compared to the existing FPGA training accelerator for the MNIST dataset. Therefore, the proposed accelerator is more suitable for deployment in mobile/edge nodes compared to the existing software and hardware accelerators.

**Keywords:** training accelerator; neural network; CNN; AI chip



## 1. Introduction

Deep learning is a type of machine learning based on artificial neural networks. An artificial neural network (ANN) is a neural network whose structure is modeled based on the human brain. A convolution neural network (CNN) extends the structure of an ANN by employing convolutional filters and feature map compression layers called pooling to reduce the need for a large number of weights. Recently, a great deal of research has been conducted in optimizing and modifying CNNs to target many applications such as image classification [1], object detection [2], and speech recognition [3], and good results have been reported. Such CNN techniques are already widely used in the industry, and they are expected to penetrate daily life, such as with autonomous driving [4] and home automation [5].

Recently, the development of IoT (Internet of Things) products and smartphones has been remarkable, consequently leading to a growing demand for CNN accelerators—not only for high-speed servers but also mobile/edge devices. The requirement of high accuracy and high speed in CNN applications such as image recognition, classification, and segmentation led the extensive research effort in developing high-speed CNN accelerators [6–9]. However, most accelerators are targeted for either high-speed servers for training large CNNs or low-power compact NPU (Neural Processing Unit) for the inference

operations of pretrained CNN models. The traditional accelerators for high speed tend to consume too much power for mobile and edge application.

To overcome this, many energy-efficient CNN accelerators for inference that are better suited to mobile/edge devices have been researched [10,11]. Some researchers have proposed software accelerators and compression techniques for the inference operation of deep learning models on mobile devices [12,13]. Most CNN inference accelerators download pretrained weights (parameters) that are calculated in advance by utilizing available trained datasets. Training involves more complex and resource-demanding processing, such as in the case of gradient-based learning for the image classification of MNIST [14]. For training in mobile/edge devices, therefore, using a general-purpose CPU (Central Processing Unit) or GPU (Graphics Processing Unit) is not realistic due to large size and poor energy efficiency. There have been many attempts to develop hardware accelerator architectures for the training of neural networks [13,15–19]. On-device training, in edge nodes, is getting increasingly common due to privacy concerns, improved performance-to-power ratios, and reduced bandwidths [13,17,19]. Another motivation for implementing edge nodes that are capable of training lies in their ability to perform online training (or even self-training) for improved accuracy [20,21]. Some existing architectures are not suitable for implementation in edge nodes due to their large size [15,19], multi-FPGA implementation [16], or low energy efficiency [13,19]. Therefore, there is a need for an energy-efficient learning accelerator for mobile/edge devices that can output high accuracy with fewer layers.

Despite the growing need for fast/low-power solutions for training of deep neural networks (DNNs) on mobile devices [22], only real-time inference solutions using dedicated hardware units and processors are now common. This is because the computing and memory resources are very limited to train deep learning models on mobile devices. To this end, research on mobile high-speed/low-power training hardware for DNNs is experiencing its dawn. Currently, such low power DNN training hardware is suffering from issues of poor learning accuracy. The complexity and difficulty of developing mobile training hardware have significantly increased because such hardware needs to support self-training functions such as semi-supervised [23] and unsupervised learning beyond supervised learning. The goal of this research was to act as a step towards developing a chip that has the capability of updating model weights while being deployed in a mobile environment. The update of weights was carried out to improve personalization to the user/environment, resulting in an improved accuracy.

The training process of CNNs typically consists of three steps: (1) inference (IF) (2) backpropagation (BP), and (3) weight gradient update (GU). These three steps are repeated using a large amount of data to get highly accurate weights. Some studies have reported the evaluation of embedded hardware architectures optimized for OPENCL (Open Computing Language) and neural networks (NNs) to enable learning [24]. However, it is difficult to apply such embedded hardware to mobile/edge devices, because their embedded software requires high speed CPUs that often incur high power consumption. Moreover, most of the previous low power accelerators aimed at inference [25] only implemented integer operators [26] to optimally handle quantized integer weights, so they lacked floating-point (FP) operations. However, training processes require FP operations to process a wide range of numbers. To calculate such values, the authors of [27] implemented an accelerator employing FP operators. However, it consumes excessive power in FP calculations because it needs to cover a wide dynamic range of intermediate values.

In this paper, we present an energy-efficient CNN training accelerator for mobile/edge devices. The key contributions of this paper are as follows.

1. We designed an accelerator architecture that can conduct both inference and training at the same time in a single chip.
2. We present a design and verification methodology for the accelerator architecture and chip implementation by converting a high-level CNN to a hardware structure and comparing the computation results of the hardware against the model.

3.     We designed an architecture that shares floating point units (FPUs) to reduce the power consumption and area.
4.     We designed a backpropagation architecture that can calculate the gradient of the input and weight simultaneously.
5.     We integrated convolution and pooling blocks to improve performance and reduce memory requirements.

Section 2 briefly describes the basic principles and behavior of CNNs. Section 3 discusses the design of the CNN accelerator and the improvements. Section 4 shows the experimental results of the designed accelerator, and finally Section 5 gives the conclusions of this paper.

## 2. Background

CNN is an abbreviation for a convolutional neural network and refers to a structure that has layers that perform convolution and pooling operations. The convolution operation extract the characteristics of the data. Much research is currently underway using CNN structures because they promise high accuracy for image processing and classification applications.

Figure 1 shows an example CNN structure used for classification, which includes one convolution, one pooling, and two fully-connected (FC) layers. Here, each layer serves a specific purpose in obtaining a classification result. The convolution layer extracts features to produce a feature map, and the pooling layer reduces the size of the feature map. Finally, the FC layer produces the classification result based on the features found in the feature map.

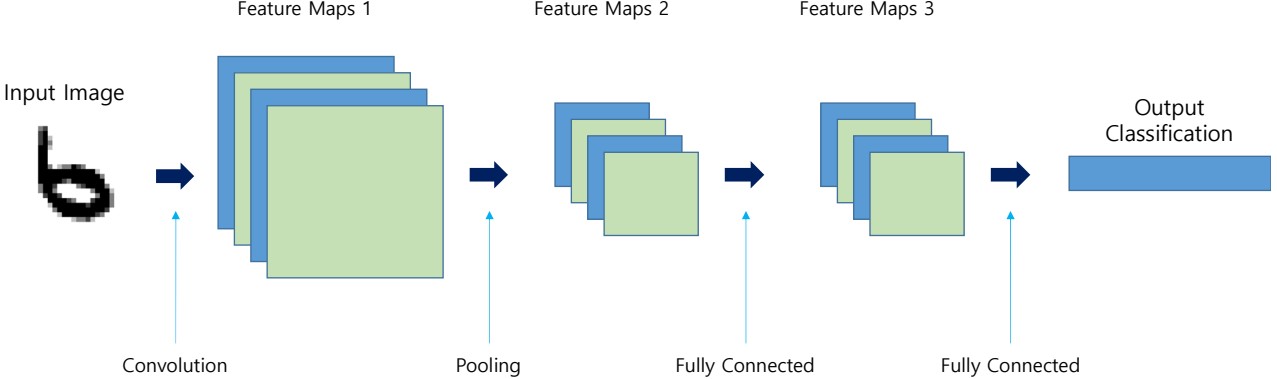

**Figure 1.** An example convolutional neural network (CNN) classification structure.

Our CNN structure targeted the MNIST dataset, with 10 classes representing handwritten digitals of 0–9. Therefore, the output data length was reduced to ten after passing through the two FC layers, which were stored in the output classification memory. Here, each data element of the memory represents the probability of corresponding class among the digits 0–9. The highest value among the ten data elements is most likely the correct answer. If, for example, the seventh value in the memory has the largest value, it is determined that digit 6 is provided as the input image. Section 2.1 discusses training in general for CNNs, Sections 2.2–2.4 discuss the operation of various layers in CNNs during inference and training, and Section 2.4 discusses the operation of the Softmax activation and cross entropy loss functions in CNNs.

### 2.1. Training in Convolutional Neural Networks

Most layers in convolutional neural networks involve manipulating the input data based on the filter/weight parameters. Training is the automatic acquisition of the optimal value of the weight and filter parameters from the training data. During training, the filter and weight parameters are tuned to achieve the desired operation. The indicators used

for the training of neural networks are called loss functions. The final goal of training is to find the weight parameters that make the outcome of the loss functions the smallest. This requires calculating the derivative (gradient) of the parameters. We repeated the process of slowly updating the values of the parameters with the calculated derivative values as a clue and explored the optimal parameters.

The training process is composed of two sub processes. Firstly, an input image is applied to the network and the image is propagated forward to obtain a classification output from the network. Secondly, the computed loss values are back-propagated to update the weights and filter parameters. This process is repeated multiple times for an available training dataset of images to reach optimal parameter values. Once a network is fully trained, only forward propagation is carried out to classify images, and the process is referred to as inference.

Computational graphs can be used to understand the process of forward and backward propagation, as shown in Figure 2. In a computational graph, the calculation that proceeds from left to right is called a forward propagation (forward), and the right to left calculation is called a backpropagation (backward). In Figure 2, the black arrow explains the forward process, and the red arrow explains the backward process.

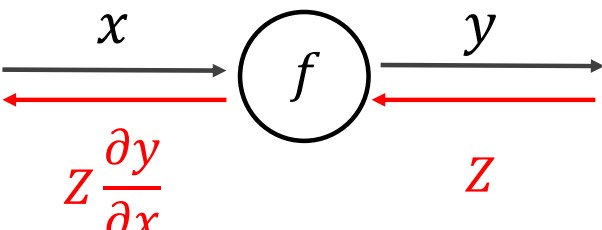

**Figure 2.** A Simple computational graph of forward and backward processes.

Firstly, in forward propagation, the input signal '*x*' is manipulated by a function $f(x)$ to obtain output signal '*y*', as represented by Equation (1). Next, in the backward computational procedure, the signal Z is multiplied by the local gradient ( $\frac{\partial y}{\partial x}$ ) of the node and then transfers it to the next (left) node. Here, the local gradient is the derivative of the expression in Equation (1), used during inference process.

$$y = f(x) \tag{1}$$

### 2.2. Fully Connected Layer

Most neural networks use an FC layer due to its generic nature, as the FC layer does not make any assumption on the nature of the input data. Each FC layers contains several neurons/nodes, each of which accepts all inputs and performs a weighted sum. Therefore, each neuron is capable of learning to identify features beyond spatial coherence. The fully connected layer, as a whole, multiplies all the weights and all the inputs to produce the result. Since all inputs and neurons meet, it is called a fully connected layer.

During inference (forward) operation, the FC layer works the same way as matrix multiplication. The output result is produced by multiplying the input *X* and weight *W* matrices, as shown in Figure 3a. Assuming an FC layer with ten inputs and ten neurons (outputs), the required weight matrix size is 10 × 10 (100 weight parameters).

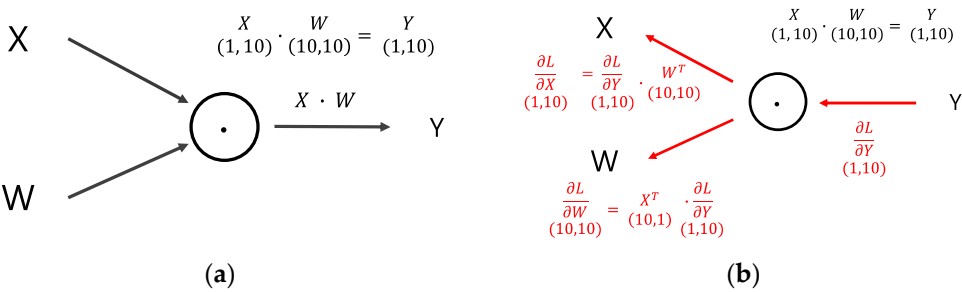

**Figure 3.** Operation of fully connected layer (**a**) in the inference process and (**b**) in the training process.

Figure 3b shows the backward process of the FC layer, where the produced loss value is *L*. It is also convenient to represent the backward process using matrix multiplication, as expressed by Equations (2) and (3). Equation (2) is used to calculate gradient of input, while Equation (3) is used to calculate gradient of weight. Here, $W^T$ represents the transpose of weight matrix *W*. The gradient of input ($\partial L/\partial X$) is propagated backward to previous (left) layers, while the gradient of weights is used to update the weights in this layer.

$$\frac{\partial L}{\partial X} = \frac{\partial L}{\partial Y} \cdot W^T L \tag{2}$$

$$\frac{\partial L}{\partial W} = W^T \cdot \frac{\partial L}{\partial Y} \tag{3}$$

*2.3. Convolution Layer*

The convolution layer convolves the input image with a set of filters to detect the characteristics of an image. Each filter activates when its corresponding feature is found at a spatial position in the input image. As a result, a feature map that highlights spots where the corresponding feature is found in the input image is produced.

The convolution operation is performed by applying a filter to the input data. Figure 4 shows an example of a convolution operation. The convolution operation applies the filter to the input data while moving it at regular intervals (stride). In each stride, the corresponding elements of the input image and the filter are multiplied together. The intermediate products are added together and stored, as shown in Figure 4 (steps 1–9). In each step of Figure 4, the process of multiplication and accumulate (MAC) is often carried out in nine steps of single multiplication-accumulation (fused multiply–add: FMA). For example, the calculation in Figure 4-(1) calculates as shown in Equation (4):

$$1 \times 1 + 1 \times 0 + 1 \times 1 + 0 \times 0 + 1 \times 1 + 1 \times 0 + 0 \times 1 + 0 \times 0 + 1 \times 1 = 4 \tag{4}$$

The result is then stored in the corresponding place of the output memory. Performing this process in all locations completes the convolutional output operation.

In the inference process, the input image and filter is processed using the FMA. During backpropagation, the process is similar to forward propagation, which is convolution. However, during the backpropagation, the output gradient is convolved with the input image to obtain the gradient of filter/weight. The gradient of the input is used as an input value in the calculation of the next layer, and the gradient of the weight is used to update the weight/filter value of the current layer. Therefore, for the first layer, the calculation of gradient of input can be skipped. Figure 5 shows the operation to obtain the gradient of weight for the convolutional layer during backpropagation process. The backpropagated value from the pooling layer is convolved with the input image to obtain gradient of weights. When there are multiple filters, the process is repeated for each filter using the backpropagated corresponding values from the pooling layer.

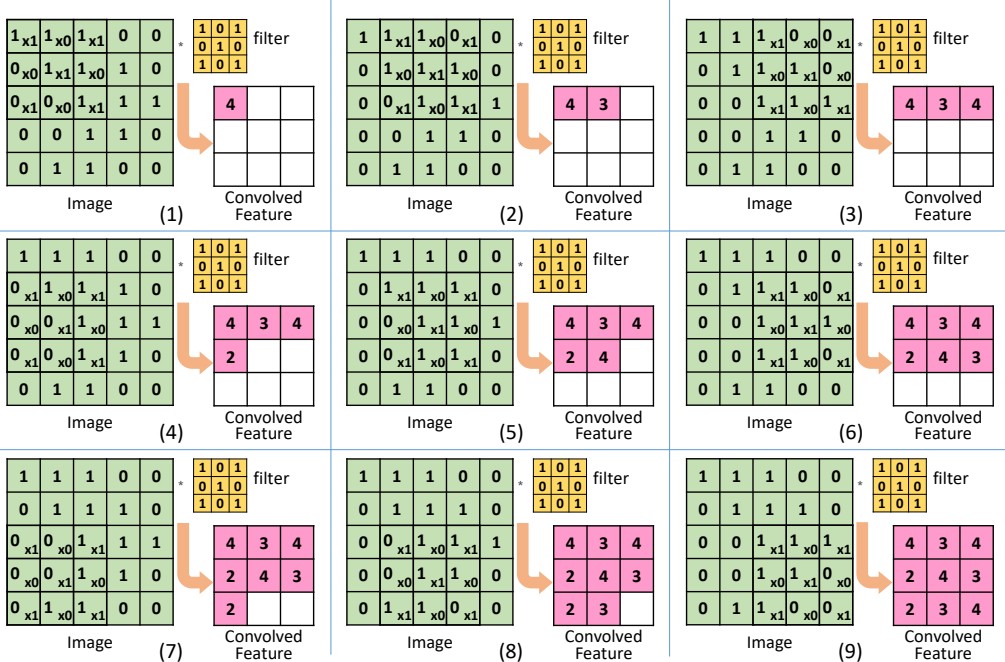

**Figure 4.** Operation of convolution layer in inference.

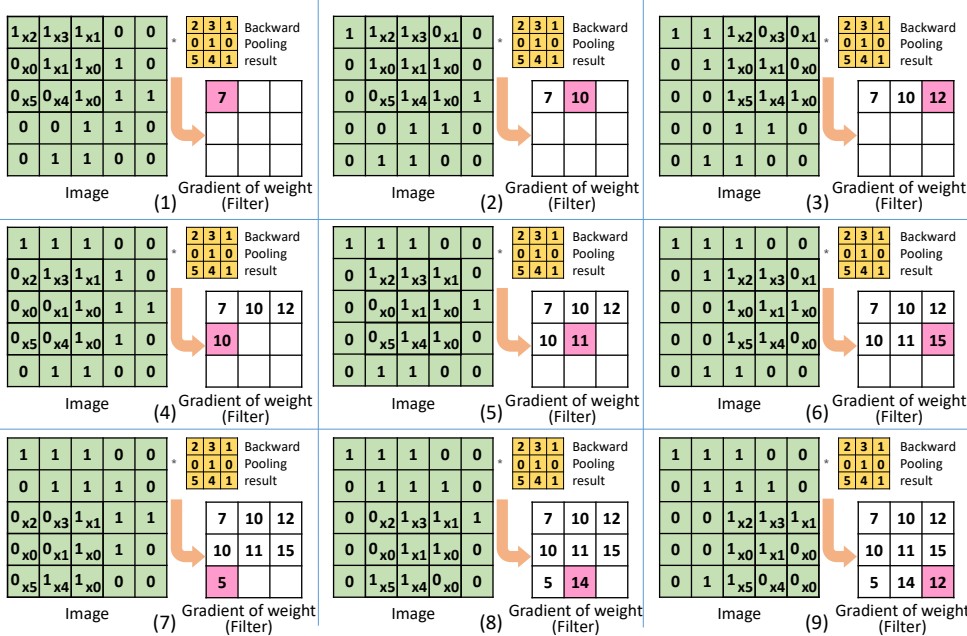

**Figure 5.** Operation of convolution layer in backpropagation.

## 2.4. Pooling Layer

A pooling layer is used to reduce the dimensions of the feature map produced by the convolutional layer. By reducing the dimensions, the memory and processing requirements are lowered in the following layers. The pooling divides the feature-map in multiple regions and then replaces the values in each region with only one representative value. The commonly used pooling methods are average pooling and max pooling, where the representative value is average or maximum value of that region, respectively. Figure 6a,b shows examples of average and max pooling on an input image, respectively. This figure uses an input image size of 4 × 4, a pooling region size of 2 × 2, and a stride of 2. As a result, the pooling output has a size of 2 × 2.

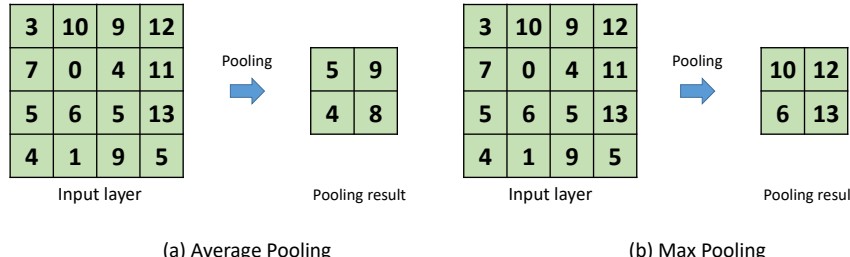

(a) Average Pooling　　　　　　　　　　　　(b) Max Pooling

**Figure 6.** Operation of average pooling (**a**) and max pooling (**b**) in inference.

During forward propagation, when the feature map passes through a pooling layer, the feature map size is reduced to a predetermined size. Conversely, in backpropagation, when the feature map passes through the pooling layer, the image size needs to be increased. The size is increased to match the size of the original feature map produced by convolution during forward propagation. To this end, the location of the largest value during the forward pooling process is stored in the argmax memory. This allows the pooling layer to remember which parts were extracted and the size of the original feature map, which can be used during backpropagation.

Figure 7a shows the operation of the pooling layer during the forward process in training. In addition to storing the pooling results for each region, it stores the positions of the maximum values extracted from each region. For example, the first region has the largest value 10 at position 1, the second region has largest value 12 at index 2, and so on. This process is repeated unless the pooling is complete. During backpropagation, the pooling layer restores to the original size of the convolution output, as shown in Figure 7b. Moreover, it places the backpropagated values at the corresponding positions stored in the argmax memory. For inference only, during forward propagation, argmax is not needed.

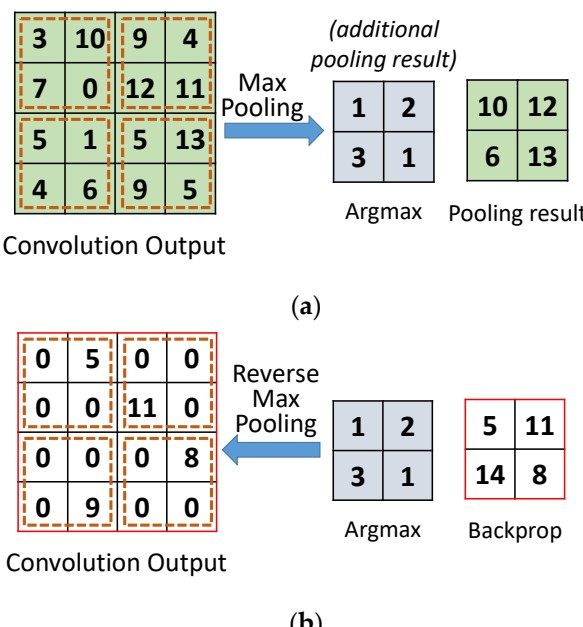

**Figure 7.** Operation of pooling layer in backpropagation: (**a**) forward propagation storing maximum value position and (**b**) backward propagation utilizing argmax memory to increase size.

*2.5. Softmax and Cross Entropy Error*

Softmax is a function used to represent the resulting value of a fully connected layer with a probability between 0 and 1. In general, the Softmax function is used for classification and is expressed as Equation (5):

$$y_k = \frac{\exp(a_k)}{\sum_{i=1}^{n} \exp(a_i)} \tag{5}$$

where $\exp(x)$ is an exponential function $e^x$, $n$ is the number neurons in the output layer, and $y_k$ is the output of the $k$-th neuron.

Usually, when training, we use a loss function as the last operation. The loss function plays an important role of searching for the optimal parameter value. The loss function is an indicator of the "bad" performance of a neural network. It indicates how well the current neural network is not processing the training data. The smaller the value of the loss function, the smaller the error. If Softmax is used as the activation function, the loss is calculated using a cross entropy error (CEE) as the loss function.

The forward and backward processes of the Softmax and CEE is shown in Figure 8. The black arrows indicate the forward process, and the red arrows indicate the backpropagation process of each function. The final value produced by the forward process is the output of the loss function, represented by $L$. The result of the Softmax and CEE function while propagating backwards is $y_1 - t_1, y_2 - t_2, y_3 - t_3 \cdots$. Since $y_1, y_2, y_3, \cdots$ represent the output of the Softmax function and $t_1, t_2, t_3, \cdots$ is the correct answer label, $y_1 - t_1, y_2 - t_2, y_3 - t_3 \cdots$ represents the error that is used in calculating gradients of inputs. Since this paper focused on updating the weight gradient, we did not use a log value in the CEE layer for the calculation of the loss value.

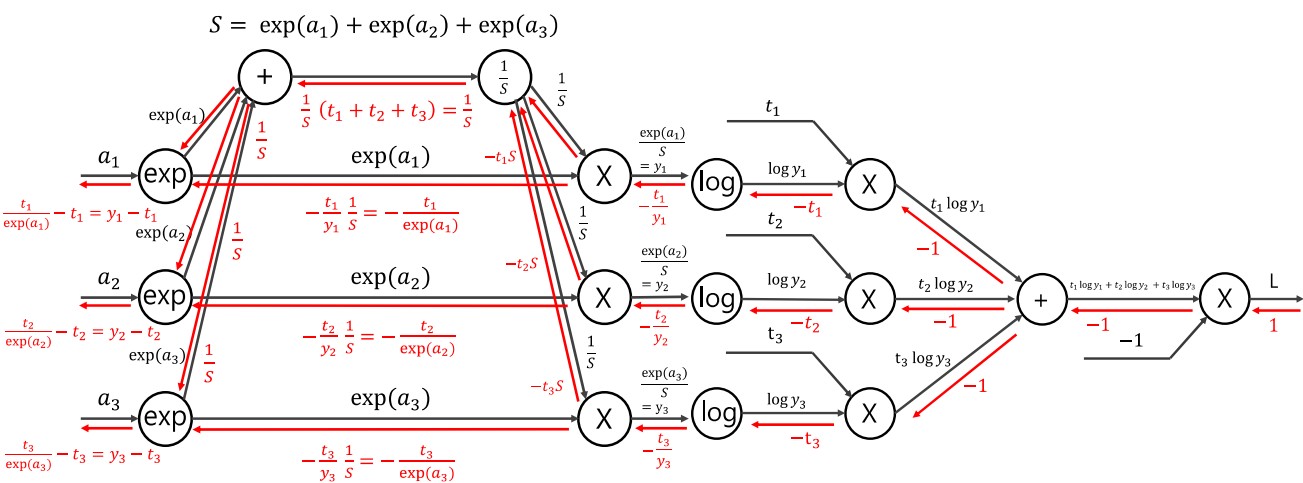

**Figure 8.** Operation of Softmax and cross entropy error function.

## 3. Architecture of Proposed Training Accelerator

The key goal of our proposed accelerator was to target the low area and energy-efficient design of mobile/edge training hardware. As the number of layers in the neural network model increases, the area and power also tend to increase. We therefore employed a network compression technique comprising layer pruning and weight quantization. As a result, we produced an optimized CNN that is shallower, with fewer filters and narrower bit-widths for weight parameters while maintaining a high accuracy. To prove the effectiveness of the proposed architecture and design methodology, we chose a small CNN and the MNIST dataset for training and validation while setting the target accuracy at 95% or higher.

Figure 9 shows an overall block diagram of the proposed accelerator. The proposed accelerator comprises five layers, which include one convolution layer, one pooling layer,

two FC layers, and one Softmax layer. The blue part indicates data path blocks for inference process (the forward propagation), while the red indicates the ones for the training process (backpropagation). The training process calculates the gradient of the input and the gradient of the weight with respect to the output loss values. The modules named dout calculate the gradient of the input, and the modules marked by dW calculate the derivatives of the weight. The gradient of dout and dW are repeatedly calculated for every layer, except for the first convolutional layer. For the first convolution layer, only dW is calculated during backpropagation, since there is no layer prior to this layer in need of dout.

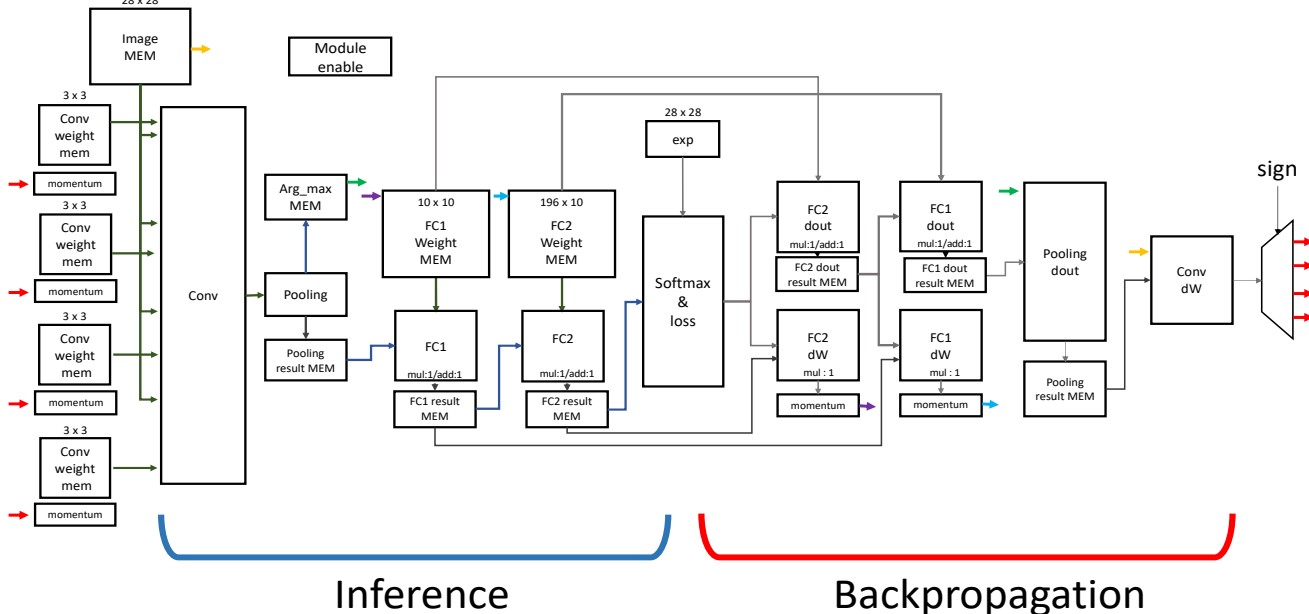

**Figure 9.** Architecture of inference and training hardware blocks. FC: fully-connected. (MEM: memory)

Each layer except convolution has a memory that stores the result of forward and backward propagation. The Softmax function employs a lookup table of 156 entries to calculate the exponential function $e^x$, for the values of $x$ in range from $-31$ to $0$. For the inference mode, the weight parameters of each layer are loaded from the outside and used, while for the training mode, the weight parameters are updated through the backpropagation operations.

### 3.1. CNN Model Optimization and Training Procedure

For the CNN optimization step, we used Tensorflow with Keras to determine the minimal number of layers in the target CNN. Figure 10 shows the accuracy and loss values changing (shown in the *Y*-axis) through a training process of 20 epochs (shown in the *X*-axis). It can be observed from Figure 10b that the loss value decreased as the epoch increased, while the accuracy increased up to the target accuracy of 95%.

If the validation accuracy satisfied the target accuracy, we continued to reduce the CNN mode by applying the network compression technique comprising layer pruning and weight quantization. We repeated this CNN model optimization and training process until the validation accuracy fell short of the target accuracy. Then, we chose the latest compressed model that satisfied the target accuracy.

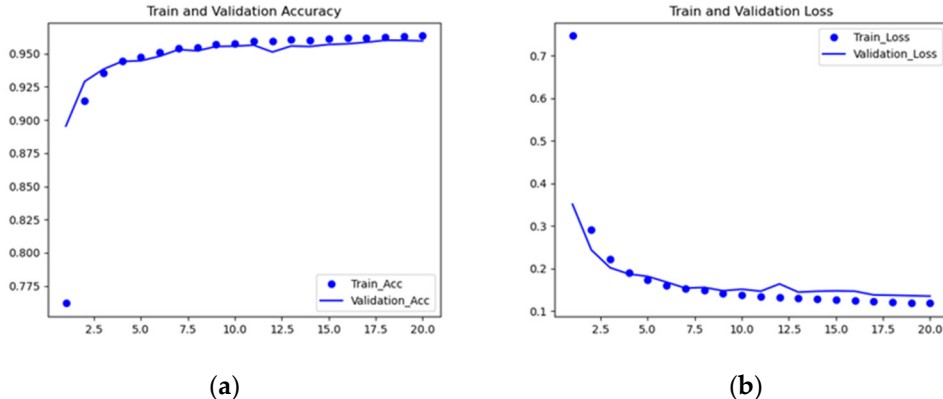

(**a**)         (**b**)

**Figure 10.** Verified accuracy and loss of TensorFlow model: (**a**) Training and validation accuracy; (**b**) training and validation loss.

In the step of CNN model optimization, we employed a high-level framework based on Tensorflow with Keras library, which is well-suited to repeated trials of network pruning and weight quantization followed by a training process.

On the other hand, this framework was not suitable for designing and verifying the accelerator hardware, since the functions of Tensorflow and Keras library are predefined and, furthermore, hidden. Therefore, once CNN optimization was finished, we convert ed the CNN model from Tensorflow/Keras to generic Python code using only NumPy library. For the remaining design steps and verification of the accelerator architecture followed by RTL (Register Transfer Level) design, we used generic Python code.

When comparing the code composed only Python language and the Keras model with the same dataset, it was confirmed that almost the same accuracy was obtained. There were some differences because the input datasets are randomly entered.

The next design step was determining the optimizer for the backpropagation of loss gradient. While the high-level model based on Tensorflow and Keras used Adam as an optimizer, the generic Python model employed a momentum optimizer, which is more hardware-friendly. The Adam optimizer requires complex arithmetic functions such as square-root and divider functions, which incur high complexity and large chip sizes. On the other hand, the momentum optimizer only requires multipliers and adders, which can be easily implemented in hardware with an RTL design. We compared the accuracy between the two CNN models—Tensorflow with Keras and generic Python—and confirmed that there was little difference in the accuracy after the complete training of long epochs. Figure 11 shows the accuracy of the CNN modeled in generic Python code using the momentum optimizer.

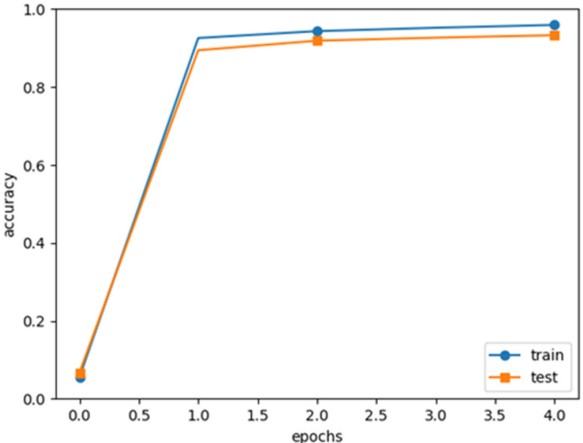

**Figure 11.** Result of native Python model training and testing.

Finally, the accelerator's architecture and RTL design were conducted, whose functionality was verified against the generic Python model in the outputs of each layer.

### 3.2. Combined Convolution and Pooling Operation

In conventional CNNs, large memories are required to store the output feature maps produced by each convolutional layer before pooling is performed. For example, with an input image size of $5 \times 5$ (25) and a filter size of $3 \times 3$ (9), the convolution layer (with stride = 1) produces a feature map size of $3 \times 3$. Moreover, the feature map size increases with the number of filters, regardless of the number channels in the input image. For instance, a convolutional layer with four $3 \times 3$ filters will produce an output of size as $3 \times 3 \times 4$. In general, when performing the pooling operation, the pooling stride is the same as pooling region size. The pooling and convolution layers, however, are different in stride and size, so they cannot be simultaneously operated. Therefore, a convolution feature map memory is required to save the results, which are used as input by the pooling layer after the convolution operation is completed.

With an increase in image size and number of filters, the feature map memory size grows to significantly impact the chip size of the accelerator. It is therefore necessary to reduce the feature map memory size for mobile/edge applications. The proposed accelerator requires an image size of $28 \times 28$ (784 values). In addition, it employs padding to prevent image size loss after convolution. After adding 1 pixel padding, the image processed by the convolutional layer has a size of $30 \times 30$ (900) pixel values.

The convolutional layer comprises four $3 \times 3$ filters and thus conventionally requires a memory of size $28 \times 28 \times 4$ (=3136 values) to store output feature maps. In order to reduce the memory size, the convolutional layer and the pooling layer are combined into a single operation block so that the two layers calculate their operations at the same time and only need memory to store the pooling result.

Figure 12 shows the operation of the proposed block that combines the convolution and pooling operations. In a conventional convolution operation, the filter moves horizontally in steps specified by stride number and then moves vertically only upon reaching the edge of the image. However, in the proposed block, the movement of the filter considers the regions defined for pooling in Section 2.3. The filter moves in a zigzag manner within a region and moves to the next region only upon completing the current region. This allows pooling to be performed on the completed region while the filter is filter is ready to move to the next region.

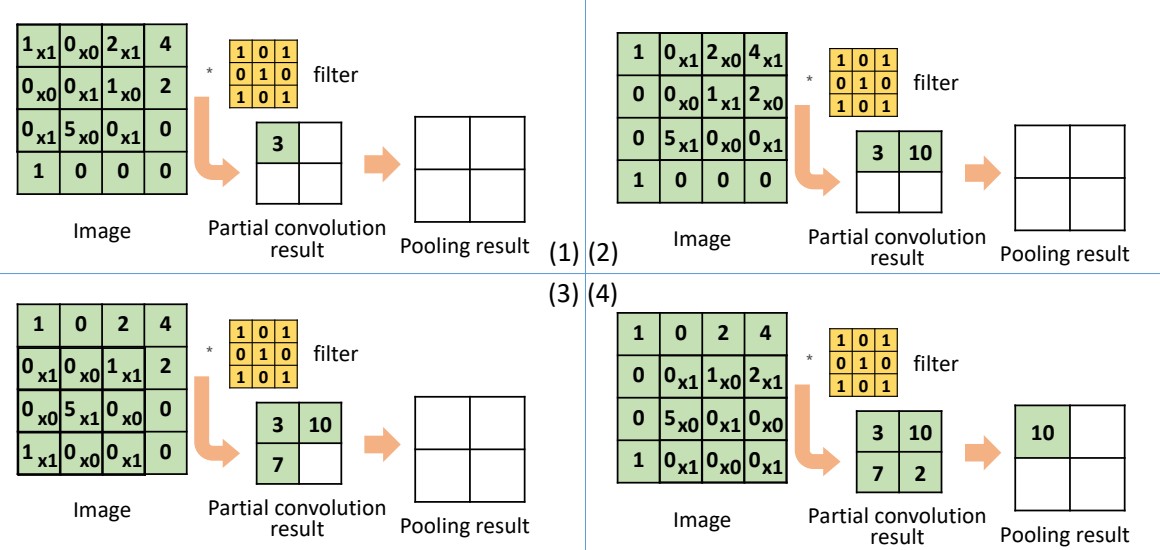

**Figure 12.** Operation order of combine convolution and pooling layer.

Figure 12 illustrates the operation of the proposed block in detail, where a small register of size equal to the pooling region, instead of the feature map memory, is used. In Figure 12 in step 1, the first MAC operation is performed, and its result is stored in the register. In step 2, the filter moves to the right and a second MAC operation is performed. In step 3, the filter moves to left down to perform the third MAC operation. In step 4, the filter moves to right and the fourth MAC operation is performed. The results of all the four MAC operations are stored in the corresponding register entries, upon which the pooling operation is performed. Similarly, the MAC and pooling operations are repeated for the next pooling regions, thus overwriting the register values.

When using the conventional/sequential method shown in Figure 12, the convolution takes 3136 clocks cycles and pooling takes an additional 196 cycles. Therefore, it takes a total of 3332 clock cycles and needs a feature map memory. On the other hand, the proposed method of Figure 12 takes only 3136 clock cycles, as the pooling operation is calculated in parallel. Moreover, the proposed method only needs a small register instead of the feature map memory, thus reducing the overall chip size.

### 3.3. Operation of Training and Inference Mode

The proposed neural network accelerator supports two modes of operation—inference and training. When trained weights are available, obtained either through training mode or loaded from external source, the inference mode is used to classify images. In inference mode, the input images are applied to the first convolution layer and the classification result is obtained from FC2 layer. This process is repeated once for a given set of images to validate the accuracy of the trained network. In the training mode, on the other hand, both forward and backward operation are involved in order to acquire trained weights. The forward operation classifies an image, as represented by probability for each class. The final probabilities are compared with one-hot coded label (true value) for the applied input image to calculate the error. The error is propagated backwards to perform change in weights/filter values. The process is repeated multiple times for a given set of images to achieve a high classification accuracy. Here, the number of images in the dataset and epoch can be adjusted.

### 3.4. Resource Sharing

For area- and energy-efficient hardware, there is a need to eliminate unused modules and reuse them. Arithmetic operations in convolution and fully connected layers involve multiplication, addition, and MAC operations in both forward and backward propagation. Moreover, the next layer idles while the current layer utilizes its arithmetic units. Taking advantage of this, the MAC and multipliers units are moved out of each layer. These modules are shared by multiple layers through the use of encoders and decoders. Figure 13 shows the complete structure of the proposed accelerator when multiple blocks share MAC and multiplier units. When running a model in software using a GPU, the priority is to run immediately so that several operations are processed at once. However, the focus of this paper was on energy efficiency and small area over speed, since the accelerator is to be deployed in energy-efficient mobile/IoT nodes.

Figure 14 illustrates the use of encoders and decoders for the resource sharing of MAC and multiplier units. Here, the multiplication and MAC units are separated from each other for the sake of speed. In backpropagation, the gradient of the input is calculated using the MAC operation, while the gradient of weight involves the multiplication operation. These two operations can be performed by reusing one module, but this consumes more time. Therefore, we added one multiplication module (in addition to the MAC unit) to perform the gradient of weight and gradient of input operations in parallel during backpropagation.

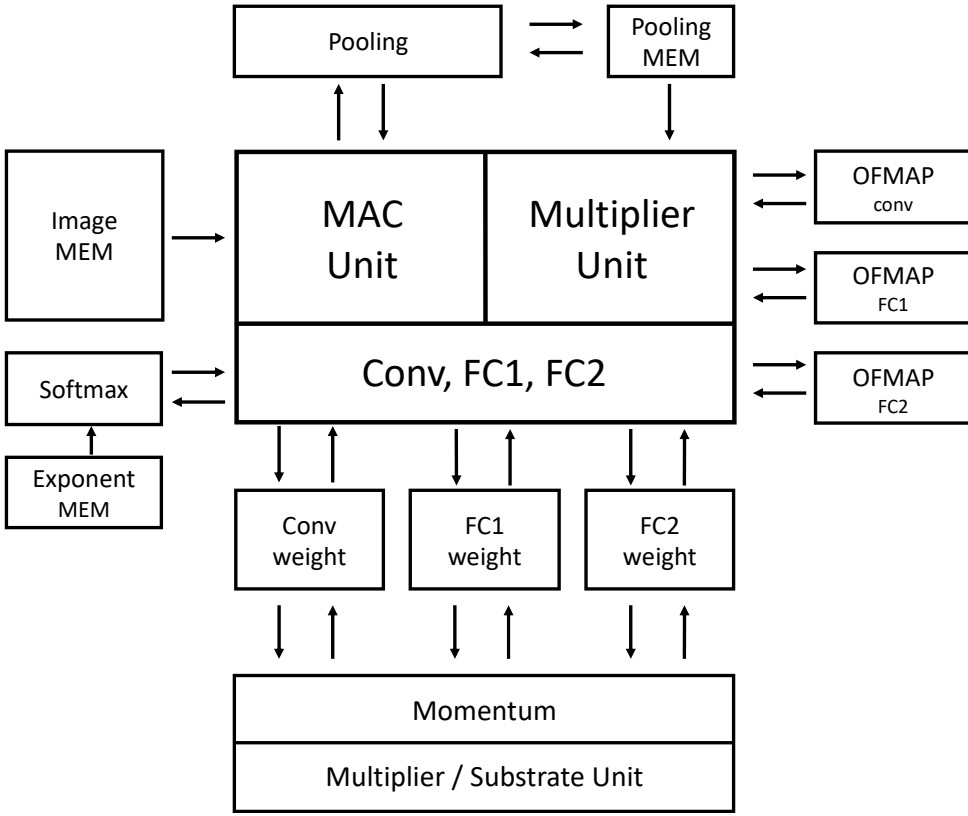

**Figure 13.** Hardware resource sharing by various layers in the neural network. MAC: multiplication and accumulate.

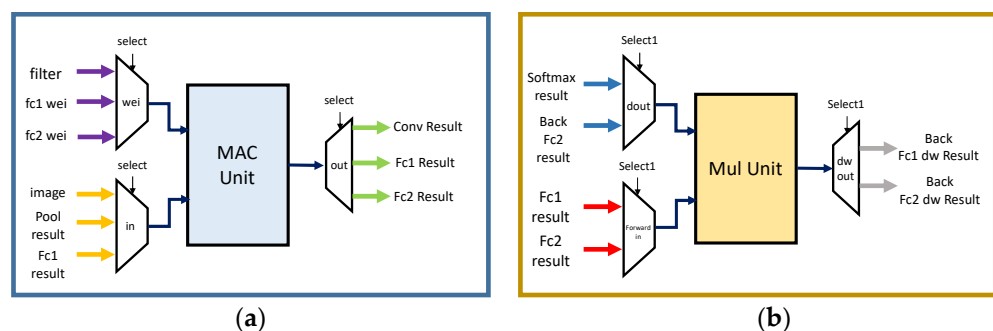

(**a**)  (**b**)

**Figure 14.** Resource sharing of operations for forward and backward propagation. (**a**) Multiplier and add operation for convolution and FC; (**b**) multiplier for calculate gradient of weight in FC.

The circuit proposed in this paper calculates the gradient by reusing the MAC and multiplier/subtractor units, which offers ease in scalability. If more (or larger) layers are needed for the training/inference of more complex images, the memory and MAC/multiplier/subtractor units can be further reused for calculation. The control logic can conveniently reallocate the memory and timing for a differently-sized network to seamlessly operate. For a larger network, a larger size memory is needed for storing the parameters and intermediate results. Since the current implementation uses on-chip memories, if a network is too large to fit, resynthesis is needed.

## 4. Implementation and Evaluation

We implemented the proposed CNN training accelerators for mobile and edge computing. We designed the architecture and synthesizable RTL code using Verilog. Then, we used the RTL code to implement silicon chip based on TSMC 65 nm CMOS process

using Synopsys's Design Compiler and IC Complier, targeting a clock frequency of 100 MHz. To ensure the correctness of the synthesis result, post-synthesis simulations were performed. Since the silicon chip was not ready for measurement yet, we performed FPGA implementation to verify correctness of the design. For the silicon chip, the power consumption and chip area were measured using Synopsys tools.

Firstly, we compared the inference accuracy of MNIST classification between (1) the Python model running on a GPU and (2) the proposed accelerator using the Verilog model on an FPGA. For this test, we used the same trained weights in both environments and chose 2200 images from the MNIST test dataset. The trained weights were obtained by training the native Python to provide an accuracy of 95% or higher. Figure 15 shows the test setup using a Xilinx ZCU104 FPGA board and Raspberry Pi 4. The Raspberry Pi was used to read/write weights, images, and results from the implemented CNN model via SPI (Serial Peripheral Interface Bus). Moreover, the Raspberry Pi computed the accuracy based on the results obtained from the implemented model on the FPGA.

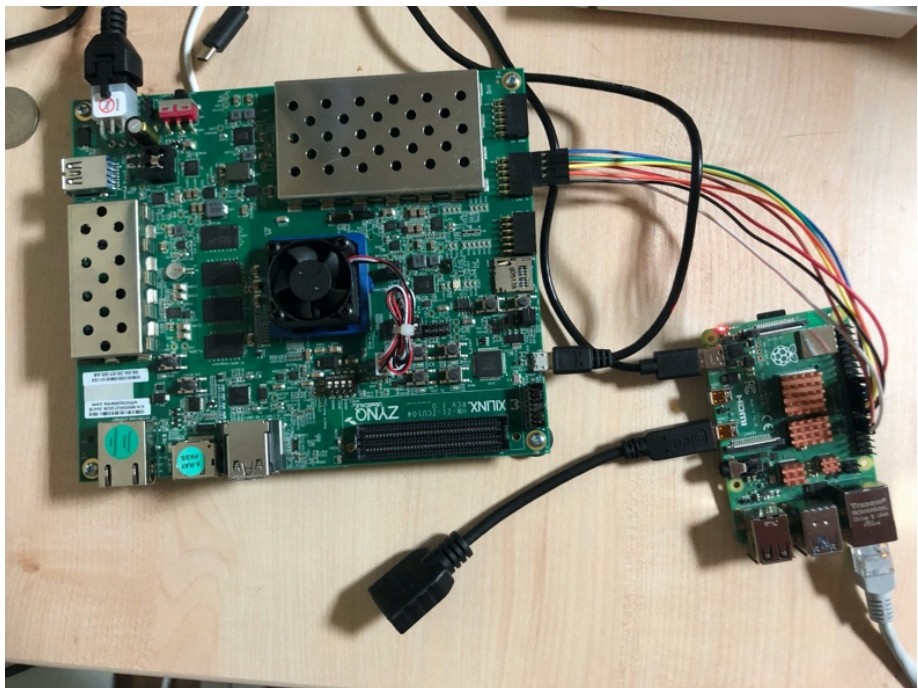

**Figure 15.** Environment of simulation using hardware (FPGA board).

Table 1 shows the evaluation results, revealing an inference accuracy of 96.05% and 2113 out of 2200 correct predictions for both the test environments/platforms. This evaluation ensured that the functionality of the proposed accelerator implementation was correct, and its floating-point arithmetic operators were also highly accurate.

**Table 1.** Comparison of Python and RTL test accuracy in inference.

|  | **Python Model** | **Verilog RTL/Hardware Model** |
|---|---|---|
| Accuracy | 96.05% | 96.05% |
| Number of Correct prediction image | 2113 | 2113 |

Table 2 shows the results of inference after separately training in Python and Verilog simulation using 2200 training images. Initially, partially trained weights were obtained from Python, providing a test accuracy of 93.5% for 1000 test images. Afterwards, these partially trained weights were again provided to Python and Verilog for further training. After separately training in Python and Verilog models, the test accuracy was compared, as

shown in Table 2. The Python and Verilog trained models showed similar test accuracies of 95% and 95.1%, respectively. This suggested that even if complete training was separately carried out in Python and Verilog, the deviation was expected to be small.

**Table 2.** Comparison of Python and RTL test accuracy for training and inference.

|  | **Python Model** | **Verilog RTL Model** |
|---|---|---|
|  | **Initial Accuracy (Weight)** | **93.5%** |
| Number of Training Data | 2200 | 2200 |
| Number of Inference Data | 1000 | 1000 |
| Accuracy | 95% | 95.1% |
| Number of Correct Prediction Images | 950 | 951 |

To highlight the benefit of resource sharing, in terms of area and energy consumption, we implemented and compared two accelerators using Synopsys: (1) an accelerator with dedicated resources and (2) an accelerator with shared resources. Tables 3 and 4 compare the chip area and power consumption of the two architectures. From Tables 3 and 4, it can be observed that the architecture with shared resources provided the best results.

**Table 3.** Comparison in terms of power for dedicated vs. shared resources architectures.

| **Accelerator with Dedicated Resource** | | **Accelerator with Shared Resources** | |
|---|---|---|---|
| **Logic** | **Power (mW)** | **Logic** | **Power (mW)** |
| Combinational | 26.069 (79.71%) | register | 22.8959 (78.88%) |
| Clock network | 0 | sequential | 0 |
| Register | 6.636 (20.29%) | combinational | 6.1314 (21.12%) |
| Total | 32.7041 mW | Total | 29.0273 mW |

**Table 4.** Comparison in terms of area for dedicated vs. shared resources architectures.

| **Accelerator with Dedicated Resource** | | **Accelerator with Shared Resources** | |
|---|---|---|---|
| **Logic** | **Area ($\mu m^2$)** | **Logic** | **Area ($\mu m^2$)** |
| Combinational | 1,672,173.38 | Combinational | 1,570,811.06 |
| Buf/Inv | 36,890.2 | Buf/Inv | 35,927.28 |
| Non-combinational | 173,657.77 | Non-combinational | 1,726,553.89 |
| Total | 3,408,531.16 | Total | 3,297,364.95 |

Table 5 compares the energy efficiency of three hardware systems with different configurations:

(1) GPU-based PC running a CNN model (using Keras/Tensorflow framework).
(2) Proposed accelerator with dedicated resources.
(3) Proposed accelerator with shared resources.

While the precision employed by the GPU-based PC was a 64-bit floating point, the precision implemented in the two proposed accelerators was a 32-bit floating point. Since the number of bits was different, a minor difference in the calculation result could be expected. However, since the order of the magnitudes of the weight values was the same in training, it could be confirmed that the weight values were updated by making the best use of each feature. In Table 5, the energy consumption was calculated by multiplying the average power consumption of the training process multiplied by the projected time span for five epochs of the complete 50,000 training images of MNIST dataset. For the GPU-based PC, the average power consumption and the overall training time are reported. During training on PC, a smart Wi-Fi plug [28] was used to measure the power of the PC box (excluding any peripherals). The smart plug communicated the voltages, current,

power, and energy consumed to the app on a smartphone over Wi-Fi, as shown in Figure 16. To calculate the energy consumed by the PC, the time duration of training was multiplied by the average power consumption. On the other hand, for the proposed accelerators, the average power consumption was estimated by Synopsys' Design Complier, and the overall training time was estimated by the Vivado Verilog simulator. Our experiment, reported in Table 5, demonstrated that the proposed accelerator with shared resources consumes only 0.21% of the energy consumed by the GPU-based PC.

**Table 5.** Comparison in terms of energy consumption. FP: floating point.

| | GPU-Based PC * | Accelerator with Dedicated Resource | Accelerator with Shared Resources |
|---|---|---|---|
| Bit-precision | FP64 | FP32 | FP32 |
| Power | 54.6 W | 32.7041 mW | 29.0273 mW |
| Time | 101 s | 393.21 s (10 ns (clock period) × 157,284 cycles × 50,000 images × 5 epoch) | 393.21 s (10 ns (clock period) × 157,284 cycles × 50,000 images × 5 epoch) |
| Energy | 5514.6 J | 12,859.6 mJ | 11,413.82 mJ |

* For GPU-based PC: Nvidia GeForce GT 710 with Intel(R) Core™ i5-9600KF CPU @ 3.70 GHz.

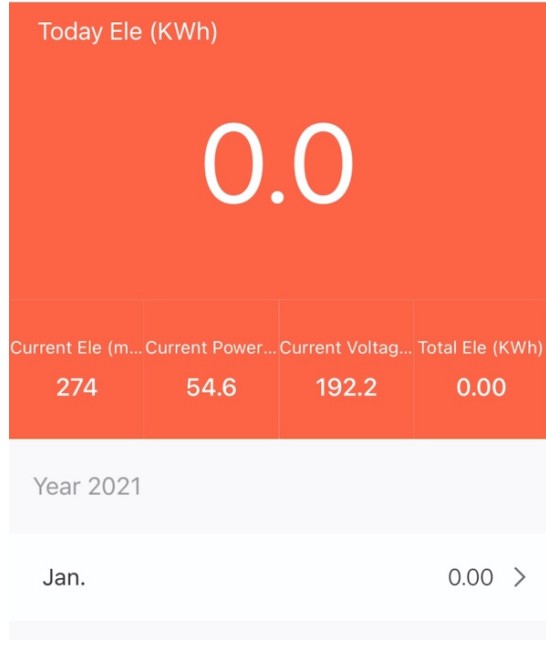

**Figure 16.** The method of average power consumption using a PC.

In order to compare with the existing FPGA accelerators [13,15,29], we obtained the FPGA resource usage and power of the proposed architecture, as shown in Table 6. For a fair comparison among accelerators targeting various sized networks, we calculated the scaling factor, which was the ratio of operations in an existing work to our work. The results reveal a normalized energy consumption of 17.40 μJ by our design, which was lower compared to the results of [15,29]. The training accelerator in [13] employed batch size of 10 to reduce latency to 363.80 μs per image, resulting in a reduced energy consumption. However, using a batch size of 1 (same and ours) would have resulted in much higher energy consumption in [13] due to the added latency of frequent weight updates to DRAM (Dynamic Random Access Memory). Therefore, the low effective normalized energy per image makes the proposed architecture suitable candidate for IoT/mobile deployments.

**Table 6.** Comparison of resource usage, execution time, average execution power, and normalized energy consumption with the existing works.

| | [15] | [29] | [13] | Proposed |
|---|---|---|---|---|
| Precision | FP 32 | FP | Fixed 16 | FP 32 |
| Training dataset | MNIST | CIFAR-10 | CIFAR-10 | MNIST |
| Device | Maxeler MPC-X | Xilinx ZU19EG | Intel Stratix 10 | Xilinx XCZU7EV |
| LUT | 69,510 | 329,288 | ALM = 20,800 | 169,143 |
| FF | 87,580 | 466,047 | | 219,372 |
| DSP | 23 | 1500 | 1699 | 12 |
| BRAM | 510 | 174 | 10.6 | 304 |
| Operations (OPs) | 14,149,798 | 74,430,000 | 59,299,400 | 114,824 |
| Scaling Factor (OPs/OPs) | 123.23 | 648.21 | 516.44 | 1.00 |
| Time Per Image (μs) | 355.00 | 864.26 | 363.80 | 26.17 |
| Power (W) | 27.30 | 14.24 [a] | 20.63 | 0.67 [a] |
| Energy Per Image (μJ) | 9691.50 | 12,307.05 | 7506.25 | 17.40 |
| Normalized Energy (μJ) [b] | 78.65 | 18.99 | 14.53 | 17.40 |

[a] Provided by Xilinx Vivado. [b] Normalized energy is energy per image/scaling factor or OPs/(OPs/s/W)/scaling factor.

For the measurements of Table 6, a Raspberry Pi was used for delivering images/weights and reading results, which consumed an additional 0.808 W of power. Upon including the power of the external device (Raspberry Pi), the consumed power and energy values increased to 1.478 W and 38.67 μJ, respectively. In real deployment, however, we plan to use an on-chip RISC-V core, which would have a much lower power consumption than the Raspberry Pi.

The layout of the implemented test core, using TSMC 65 nm CMOS process and generated by a Synopsys IC Complier, is shown in Figure 17. The test core included the accelerator with shared resources and an off-chip interface for writing image/weights and reading produced results. The core occupied an area of $2.246 \times 2.246$ mm$^2$.

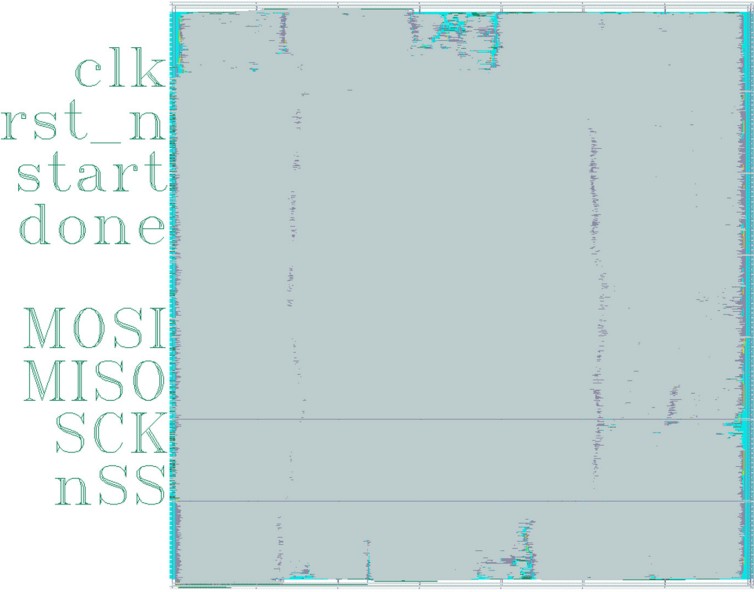

**Figure 17.** Layout of the accelerator with shared resources.

## 5. Conclusions

This paper proposed a single-chip training and inference accelerator for use in mobile/edge and IoT devices. The inference and training functions of the accelerator were evaluated for MNIST handwritten digits 0–9 on an FPGA. The accelerator locally stores weights and only needs external communication for acquiring input images and reporting results. The architecture has the benefit of a reduced energy consumption compared to a

GPU-based accelerator running on a PC, and it demonstrated similar training and inference accuracies. Additionally, the proposed low-power accelerator has the capability for continuous self-training while deployed, thus ensuring a high accuracy of prediction. The accelerator promises huge potential in the growing number IoT edge devices by offering training capabilities at a fraction of the power consumption of a model running on a GPU. To further reduce area and energy consumption, pruning and quantization down to 16-bit or 8-bit floating points may be used for calculation. To use the proposed accelerator for various applications, we plan to develop a flexible architecture, where the same hardware can be used to accelerate networks of different depths, in the future.

**Author Contributions:** Conceptualization, J.H. and H.K.; methodology, J.H., S.A. and H.K.; software, J.H. and T.L.; validation, J.H. and S.A.; formal analysis, J.H. and S.A.; investigation, J.H. and T.L.; resources, T.L. and S.A., Writing—original draft preparation, J.H.; Writing—review and editing, S.A. and H.K.; supervision, H.K.; project administration, H.K.; funding acquisition, H.K. All authors have read and agreed to the published version of the manuscript.

**Funding:** This work was supported by IITP grant (No. 2020-0-01304), Development of Self-learnable Mobile Recursive Neural Network Processor Technology Project, and also supported by the Grand Information Technology Research Center support program (IITP-2020-0-01462) supervised by the IITP and funded by the MSIT (Ministry of Science and ICT), Korean government. It was also supported by Industry coupled IoT Semiconductor System Convergence Nurturing Center under System Semiconductor Convergence Specialist Nurturing Project funded by the National Research Foundation (NRF) of Korea (2020M3H2A107678611).

**Conflicts of Interest:** The authors declare no conflict of interest.

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
