# Peer review of "Design of Power-Efficient Training Accelerator for Convolution Neural Networks"

_electronics, doi:10.3390/electronics10070787_

Round 1
Reviewer 1 Report
The authors provide an ASIC implementation of a CNN-type neural network, with the possibility of including training (in addition to inference). It seems that the main contributions of this article are precisely the incorporation of training and the fact that it has a very low power consumption, which makes it a very energy-efficient solution.
The implementation of any network and being done with Verilog is very meritorious. Even more so if it includes built-in training and I think that the effort of verification through prototyping that they have done with the incorporation of XILINX FPGA also has an extraordinary merit for someone who reviews them who has spent many years designing and verifying digital circuits.
Obviously, I know what happens when such a complex development has been carried out and an attempt is made to publish it, which I will summarise in two questions:
1. Where are the most relevant contributions of the article? and.
2. Are they justified?
Let's start with the justification. In the world of neural network implementation, it is very adventurous to pretend to provide solutions for a network that incorporates learning by means of ASIC technology in the solutions called "mobile edge devices" by the authors. I honestly do not see this kind of need (I am referring to on-line training). There are two things that the authors need to improve:
1. The references that justify it. They only provide one reference (19), which is poorly referenced in the article (it lacks a title) and reading it does not show that it is really an adequate type of justification (the authors talk about offline training by adding samples acquired in their own inference system, but obviously the acquisition of inputs). It should be borne in mind that in order to carry out training (if it is supervised) it is necessary to know the target and I find it difficult to understand (and I hope the authors will explain it to me) in what type of applications they see the need to carry out on-line training of these characteristics.
2. Any self-respecting training requires a highly flexible implementation that allows variation of the network topology as a minimum. I have not seen in the article any mention of the extent to which this network is flexible to parameter changes (more layers, more neurons per layer, different learning factors, different optimization systems). In inference, this flexibility would also be desirable.
Assuming the justification best explained and justified by the authors we now turn to the contributions that are named in the title, abstract and conclusions: the very low energy consumed by their solution.
It seems that section 4 on implementation and evaluation is the most important part to demonstrate that contribution and several questions arise for me:
1) You compare only with PC+GPU solutions. It would be interesting if you could indicate especially the GPU characteristics (since the variety of architectures and their evolution requires more details). And it is also desirable that they explain how they measured the energy consumption of this solution.
2) As for the solution provided by the authors, I would still ask for more details. Using the power consumption values provided by the synopsys tools seems to me to be unrepresentative unless several isolated ASICs were compared. But for the comparison provided by the authors, it is necessary to compute and measure or predict the power consumption of the infrastructure needed to deliver the samples and parameters to the ASIC and to obtain the network results from the ASIC. Even if we focus on the results provided by Synopsys, it would be necessary to detail which toggle rate has been used and if they have made any assessment of it through gate-level simulation.
Finally, there are some details that show that the authors do not know what to contribute and how to demonstrate these contributions. I would like them to explain to me what the end of page 15 and the beginning of page 16 (including table 6) contributes to the article, in which they compare the prototype used to verify their solution (I imagine a Xilinx FPGA) with a solution (when there are hundreds and hundreds of solutions) that implements a similar network with FPGAs. This comparison does not make sense and if it did, the authors should review and update the references with other much more current and representative ones.
Reviewer 2 Report
The topic addressed in the manuscript is potentially interesting and the manuscript contains some practical meanings, however, there are some issues which should be addressed by the authors:
1) The "Abstract" and "Introduction" sections can be made much more impressive by highlighting your contributions. The contribution of the study should be explained simply and clearly.
2) The authors should clearly emphasize the contribution of the study. Please note that the up-to-date of references will contribute to the up-to-date of your manuscript. The studies (https://doi.org/10.3390/s21041038) can be used to explain your method and optimization in the study or to indicate the contribution in the Introduction section.
3) Conclusion section should be rearranged. According to the topic of the paper, the authors may propose some interesting problem as future work in conclusion.
This study may be proposed for publication if it is addressed in the specified problems.
Round 2
Reviewer 1 Report
I appreciate the effort to improve the article, especially with the time constraints that usually accompany the revision process.
I still think that the energy measurement performed by the ASIC solution is weak or at least ill-conceived, but I also understand that it is not something that can be fixed in a few weeks.
If the flexibility of the solution is given by pre synthesis parameters, I foresee problems for this solution to have a future; but I think this question is not relevant to the review of this article at this time.
Reviewer 2 Report
No comments